**Data Availability Statement:** Raw sequence reads are publicly available at the European Nucleotide Archive under PRJEB35160. De-identified patient

# *Moraxella*-dominated pediatric nasopharyngeal microbiota associate with upper respiratory infection and sinusitis

Kathryn E. McCauley[1], Gregory DeMuri[2], Kole Lynch[1], Douglas W. Fadrosh[1], Clark Santee[1], Nabeetha N. Nagalingam[1], Ellen R. Wald[2], Susan V. Lynch[1]*

1 Division of Gastroenterology, Department of Medicine, University of California San Francisco, San Francisco, CA, United States of America, 2 Department of Pediatrics, University of Wisconsin School of Medicine and Public Health, Madison, WI, United States of America

* susan.lynch@ucsf.edu

## Abstract

### Background

Distinct bacterial upper airway microbiota structures have been described in pediatric populations, and relate to risk of respiratory viral infection and, exacerbations of asthma. We hypothesized that distinct nasopharyngeal (NP) microbiota structures exist in pediatric populations, relate to environmental exposures and modify risk of acute sinusitis or upper respiratory infection (URI) in children.

### Methods

Bacterial 16S rRNA profiles from nasopharyngeal swabs (n = 354) collected longitudinally over a one-year period from 58 children, aged four to seven years, were analyzed and correlated with environmental variables, URI, and sinusitis outcomes.

### Results

Variance in nasopharyngeal microbiota composition significantly related to clinical outcomes, participant characteristics and environmental exposures including dominant bacterial genus, season, daycare attendance and tobacco exposure. Four distinct nasopharyngeal microbiota structures (Cluster I-IV) were evident and differed with respect to URI and sinusitis outcomes. These clusters were characteristically either dominated by *Moraxella* with sparse underlying taxa (Cluster I), comprised of a non-dominated, diverse microbiota (Cluster II), dominated by *Alloiococcus/Corynebacterium* (Cluster III), or by *Haemophilus* (Cluster IV). Cluster I was associated with increased risk of URI and sinusitis (RR = 1.18, p = 0.046; RR = 1.25, p = 0.009, respectively) in the population studied.

### Conclusion

In a pediatric population, URI and sinusitis associate with the presence of *Moraxella*-dominated NP microbiota.

data and the final analysis OTU table are available from: https://github.com/lynchlab-ucsf/SinusitisMicrobiome.

**Funding:** This work was funded by NIH R01 AI097172: "Sinusitis in Children and the Nasopharyngeal Microbiome".

**Competing interests:** S.V. Lynch is a board member and reports personal consulting fees from Siolta Therapeutics outside the submitted work; has a patent "Reductive prodrug cancer chemotherapy (Stan449-PRV)" issued, a patent "Combination antibiotic and antibody therapy for the treatment of Pseudomonas aeruginosa infection (WO2010091189A1)" with royalties paid by KaloBios, a patent "Therapeutic microbial consortium" and "Methods and Compositions Relating to Epoxide Hydrolase genes" licensed to Siolta Therapeutics Inc., a patent "Systems and methods for detecting antibiotic resistance (WO2012027302A3)" issued, a patent "Nitroreductase enzymes (US7687474B2)" issued, a patent "Sinusitis diagnostics and treatments (WO2013155370A1)" licensed by Reflourish, and a patent "Methods and systems for phylogenetic analysis (US20120264637A1)" issued. This does not alter our adherence to PLOS ONE policies on sharing data and materials. All other authors declare no relevant conflicts of interest.

# Introduction

Viral upper respiratory infection (URI) is the most common illness for which children present to their primary care provider. Approximately 5–10% of viral URIs are complicated by acute bacterial sinusitis [1]. Sinusitis results in over $5.8 billion in health care expenditures in the United States annually, of which $1.8 billion are spent on children under the age of 13 years [2]. Acute bacterial sinusitis is usually preceded by a viral respiratory infection. Viral URI causes mucosal inflammation within the nose and nasopharynx that promotes obstruction of the sinus ostia. Virus-induced proliferation of pathogenic bacteria in the nasopharynx can set the stage for the development of complications such as acute sinusitis and acute otitis media [1,3]. Our previous investigations described characteristics of the nasopharyngeal microbiome in a longitudinal cohort of children during a healthy state and during URI [4]. In addition, virus identification and bacterial colonization was assessed when the children were asymptomatic and at the onset of uncomplicated URIs [1,3,4].

To date, only one external study has examined the relationship between acute bacterial sinusitis and nasopharyngeal microbiota, utilizing culture-based techniques to identify pathogens in upper airways [5]. With the advent of next-generation sequencing, understanding entire community structures may allow us to understand the heterogeneity previously observed.

Here we focus on the sub-population of children with acute bacterial sinusitis and describe and compare the nasopharyngeal microbiota in children with uncomplicated URI and those whose URIs become complicated by the development of sinusitis. We test the hypothesis that specific microbiota communities exist in these children and that these assemblages and their environmental drivers are associated with increased risk of URIs and acute sinusitis. Furthermore, we identified specific bacteria within these microbiota that may influence the occurrence of sinusitis and URI in these children.

# Methods

## Enrollment and inclusion criteria

Healthy children ages 48–96 months were recruited from February 2012 to February 2013 from two primary care pediatric practices during well-child visits in Madison, WI and followed for one year as previously described [1,3] (S1 Fig). Briefly, children were excluded if they had an underlying condition reported by the parent or noted in the medical record that was likely to alter the natural history of URI, including asthma, congenital or acquired immunodeficiency, craniofacial abnormalities, cystic fibrosis, allergic rhinitis or a history of chronic sinusitis. Children were also excluded if they had plans to move out of the area for the study duration. Demographic information, including maternal education, tobacco exposure, animal exposure, and daycare attendance were ascertained through questionnaires completed by the child's guardian. Written, informed consent was secured from legal guardians and assent was obtained from children ≥7 years of age. The study was approved by the University of Wisconsin Institutional Review Board. Subjects received a small stipend for participation. Demographic and past medical history information was obtained at the initial study visit.

## Power and sample size calculations

To assess the effects of viral presence on progression to acute bacterial sinusitis, the type and number of viruses detected, the season, and the interaction of season and virus present and demographic variables we fit a generalized linear model (GLM) to the URI episode data (1,500 observations on 300 children, 0 = non progressing, 1 = progressing) using child as the main

sampling unit and assuming a binomial distribution for the data. The average number of URI episodes per child is typically 5 per year [6]. GLM accounts for the potential correlation among multiple URI episodes in each child. For predictors which take values of yes or no (such as virus type), we calculated that our study had 80% power to detect the difference between the percent of a particular virus type equal to 30% for the acute bacterial sinusitis group and 45% for the control group. We also tested whether rhinovirus is the most common viral URI to predispose to acute bacterial sinusitis using a Chi-squared test for goodness of fit. We calculated that if rhinovirus accounts for 40% of the URIs leading to sinusitis and the second most common leads to 20% or less our study would have 80% power to conclude that rhinovirus is the most common viral antecedent to acute bacterial sinusitis.

Logistic regression was used to assess the effect of bacterial richness, evenness, and diversity, relative abundance of specific important pathogens, presence of sinopathogens, the demographic variables (including age, gender, attendance at daycare, family or personal history of asthma, pets, etc) and season on progression to acute bacterial sinusitis. A term was included to account for the paired structure of the data. For each of the continuous predictors, a difference in the mean values between the acute bacterial sinusitis group and the control group equal to 50% of the variability of that predictor would provide greater than 80% power to find a significant effect of the predictor. For predictors which take values of yes or no (such as presence of a sinopathogen), we calculated that our study would have 80% power to detect the difference between the percent of sinopathogens present equal to 55% for the acute bacterial sinusitis group and 30% for the control group.

## Classification of respiratory episodes

Each respiratory episode was classified as either an uncomplicated URI or sinusitis. The diagnosis of sinusitis was based on one of the following clinical criteria: 1) persistent symptoms—nasal discharge or cough or both, that lasted more than 10 days and were not improving (symptom score at 10 days $\geq$ 50% of highest score), or 2) worsening symptoms—sudden renewal of respiratory symptoms (nasal discharge or cough) or fever after apparent improvement usually beyond the 6[th] day of illness [7].

## Procedures

A flocked swab was placed into the nasopharynx and rotated for 10 seconds. The swab stick was cut off with sterile scissors and the swab placed into sterile DNAase/RNase-free cryovials containing 2 ml of RNALater (Ambion). Samples were collected at the University of Wisconsin, Madison, and stored at 4°C for 24 h to permit preservative to penetrate cells, prior to freezing at −80°C and then shipped in batches on dry ice to the University of California San Francisco for microbiota analysis.

Nasal samples were obtained at entry and during four surveillance visits (February, April, September and December) when children were asymptomatic as verified by the study nurses. Parents were instructed to call the study nurse at the first sign of a URI, which was defined as at least 48 hours of respiratory symptoms including nasal congestion, nasal discharge or cough. Nasal samples were obtained on day 3–4 of illness by the study nurse and a recovery sample was obtained on day 15. An additional nasal sample was obtained approximately on day 10, if and when a child was diagnosed to fulfill criteria for a diagnosis of acute sinusitis. A clinical assessment at the time of the initial visit assured that symptoms reflected infection confined to the upper respiratory tract. A symptom survey was filled out on day 3–4 and subsequently by telephone on days 7, 10 and 15 [8].

Briefly, the survey inquired about the presence of fever, nasal discharge, nasal congestion, cough, headache, irritability, facial pain, facial swelling, activity, sleep and impaired appetite. If a particular symptom was present initially, a score of 2 was assigned. If it was absent the score was 0. If a symptom became more severe, less severe, or stayed the same during the observation period, +1, -1 or 0, respectively, was added to the previous score for each symptom. The respiratory illness was considered resolved if the total score was $\leq 2$ (reflecting insignificant residual symptoms).

## Sequencing methods

A total of 477 nasopharyngeal swabs, and 10 negative controls (phosphate-buffered saline) were processed for 16S rRNA bacterial community profiling using the AllPrep (Qiagen, CA) protocol [4]. The variable region 4 (V4) region was amplified and quantified using 515F and 806R primers and a previously described protocol [9]. Amplicons were quantified using the Qubit HS dsDNA kit (Invitrogen). Of the 477 samples processed, 103 did not produce sufficient DNA for 16S rRNA amplification (<2ng/uL), resulting in 374 samples with sufficient amplicon to be included on a paired end Illumina NextSeq 500 sequence run performed as previously described [9].

Raw sequences were de-multiplexed and quality-filtered to remove low quality sequences. Sequences with three or more consecutive bases with a Q-score less than 30 were truncated and discarded if their length was less than 75% of the original 150bp read length. Paired-end sequences were merged using FLASH 38 v 1.2.7 and processed to produce an OTU table using USEARCH and Quantitative Insights into Microbial Ecology (QIIME) [10]. Dereplicated sequences were clustered into Operational Taxonomic Units (OTUs) using UPARSE at 97% identity, and chimeras were simultaneously removed. All quality-filtered reads were mapped back to the OTU sequences at 97% identity using UCLUST [11]. A phylogenetic tree was constructed using sequences that aligned using PYNAST [12], and any OTUs that failed to align were removed. Taxonomy was assigned using assign_taxonomy.py and the GreenGenes database (May 2013) [13].

OTUs possessing a number of reads that were less than 0.001% of the total reads were removed. OTUs present in at least half of the negative controls were removed from the dataset; among the Negative Control OTUs that remained, the maximum number of reads from each OTU in the negative controls was subtracted from all samples, and any resulting negative numbers (indicating fewer reads in the sample than in the negative control) were replaced by zeroes. Alpha rarefaction curves of observed species and phylogenetic diversity assisted in determining three candidate sequence read depths. For each candidate rarefying depth, the OTU table was multiply-rarefied, a process in which 100 OTU tables of the chosen rarefying depth are generated, and the sample profile located at the center of a Euclidean distance matrix for each sample was considered to be the representative profile [9]. The three candidate rarefying depths were then compared using Procrustes (transform_coordinate_matrices.py) in QIIME. Rarefying the dataset at 37,692 reads per sample resulted in 354 high-quality microbiota profiles available for analysis.

The University of Wisconsin provided de-identified patient and sample-specific data for microbiota analyses upon completion of 16S rRNA profiling by the University of California San Francisco (UCSF). Patients provided between 1 and 21 samples with a high-quality microbiota profile, with a median of 5 samples per participant. Raw sequence reads are publicly available at the European Nucleotide Archive under PRJEB35160. De-identified patient data, analysis datasets, and code are available from: https://github.com/lynchlab-ucsf/ SinusitisMicrobiome.

## Virus identification

Viral identification was performed on nasal samples by multiplex polymerase chain reaction (PCR; Respiratory Multicode Assay [EraGen Biosciences] or Respiratory Viral Panel [Luminex]) to test for the following viruses: respiratory syncytial virus (RSV; groups A and B), rhinovirus (RV; approximately 160 known types), parainfluenza (1, 2, 3, 4a and 4b), influenza (A, B and C), adenovirus (B, C and E), coronavirus (229E, NL63, OC43, HK, and severe acute respiratory syndrome), enterovirus, human bocavirus, and human metapneumovirus (A and B). Nasal specimens were also analyzed by means of partial sequencing to determine which RV types were present and differentiate closely related enterovirus from RV [14]. Resulting data was summarized by summing the total number of viruses present within each sample.

## Participant metadata

Participant demographics and early life exposures were ascertained through questionnaires taken at entry into the study, which included questions about self-reported race (Native American or Alaskan Native, Asian, Black or African American, Native Hawaiian or Other Pacific Islander, White or Caucasian, Unknown or Unreported, Other). Mixed-race participants could identify as other and were not excluded from the study. Season was defined by the date of study visit/sample collection, in which dates were aggregated into months; December, January and February were classified as Winter; March, April and May were defined as Spring; June, July and August were considered Summer, and September, October and November were defined as Fall. Tobacco exposure was ascertained through a question regarding second-hand exposure: "Is your child exposed to tobacco smoking where he/she lives?". Daycare attendance was obtained by asking "Does your child attend daycare/school for at least 10 hours a week?"

## Statistical analyses

Statistical analyses were performed using *R* v 3.6.2 and *Quantitative Insights Into Microbial Ecology* (QIIME) v1.9.1 [10]. Alpha diversity values (Chao1, Pielou's Evenness, Faith's Phylogenetic Diversity) and beta diversity distance matrices (Bray Curtis, Weighted UniFrac, Canberra, and Unweighted UniFrac) were generated in QIIME. Relationships between bacterial diversity and clinical, environmental or viral factors were assessed using Generalized Estimating Equations (GEE) (*geepack*) [15]. For analyses of beta diversity, two approaches were used. The first utilized the first principal coordinate as the dependent variable. The other applied a bootstrapped method for analyses in order to understand the proportion of variance explained by variables of interest. This second method randomly sampled independent samples 500 times, followed by PERMANOVA (*vegan*, R) [16]. The results from each of the 500 iterations were aggregated to generate a range of $R^2$ and P values, and the mean of both are reported.

Dominant genus was calculated based on the genus with the greatest number of reads in each sample and genus-level proportions were obtained using *summarize_taxa.py* in QIIME v 1.9.1. Less frequently observed dominant genera (detected in ≤10 samples) were classified as "Other". Community structures were evaluated using hierarchical clustering for all distance matrices. An average silhouette statistic from the *silhouette* package in R was used to test for goodness-of-fit of between 2 and 18 clusters, and the number of clusters with the highest silhouette statistic was chosen. Relationships between these clusters and clinical, viral, and microbiological variables were assessed using GEE with cluster category (Cluster "X" vs all other clusters) as the dependent variable.

Differential taxa were identified using a multi-model approach in which linear, Poisson, negative binomial, Tweedie, and zero-inflated negative binomial mixed effects models were applied. The Akaike Information Criterion determined the model that best fit the distribution

of data for each OTU. For the "winning" model, the estimate and p-value are reported. These models were built using the *glmmTMB* package in R, and the script is publicly available on github: lynchlab-ucsf/lab-code/SigTaxa/ManyModelScript.R.

## Results

### Environmental and demographic factors are associated with nasopharyngeal microbiota composition

Subjects in this study were primarily male, white, non-Hispanic, and possessed at least some college education (Table 1). A total of 354 longitudinally-collected nasopharyngeal samples produced a high-quality 16S rRNA bacterial profile from 58 initially-healthy subjects aged 4–7 years old. Approximately 40% of samples with a high-quality microbiota profile were obtained during periods of health (n = 141), 98 (28%) were obtained during acute URI episodes. A total of 524 taxa were identified; samples dominated by *Moraxella* were most frequently observed (175 of 354 samples, or 49%), followed by *Alloiococcus* (58, 16%), *Haemophilus* (36, 10%), *Corynebacterium* (34, 10%), *Staphylococcus* (18, 5%), and *Streptococcus* (17, 5%), reflecting frequently observed dominant genera in the upper respiratory tract.

Leveraging data generated from repeated measures, we examined factors that explained variance in the composition of the nasopharyngeal microbiota using the first principal coordinate of Canberra and Weighted UniFrac distance matrices with generalized estimating equations (GEE). Under a Canberra distance matrix, several factors related to community composition, including dominant genus (P<0.001), race (P<0.001), and visit type (P<0.001). A Weighted UniFrac related to fewer variables, though of note, season remained significant after multiple comparisons (Table 2). Several environmental exposures previously linked with protection against or development of airway disease in childhood, including dog exposure [17], daycare attendance [18] and tobacco exposure [19] were significant before but not after multiple comparisons testing.

To identify factors explaining the greatest proportion of variance in nasopharyngeal microbiota composition, we also employed a resampled PERMANOVA approach, described in

**Table 1. Demographics of subjects participating in the study.**

| | |
|---|---|
| Age range (years) | 4–6.6 |
| Gender—female (%) | 43 |
| Race (%) | |
| American Indian or Alaska Native | 1.6 |
| Asian | 3.2 |
| Black or African-American | 9.8 |
| White or Caucasian | 78.7 |
| Other | 6.6 |
| Ethnicity (%) | |
| Hispanic | 6.6 |
| Non- Hispanic | 93.4 |
| Maternal Education Level (%) | |
| Grade School | 1.6 |
| High School | 4.9 |
| Vocational/Technical | 1.6 |
| Some College | 18.0 |
| College Degree | 41.0 |
| Graduate/Professional | 32.8 |

**Table 2. Microbiological, clinical and viral factors associated with principal coordinate 1 of a Canberra and Weighted UniFrac distance matrix.**

| | Canberra | | Weighted UniFrac | |
| --- | --- | --- | --- | --- |
| Variable | P-value | FDR P-value | P-value | FDR P-value |
| Dominant Genus | <0.001 | <0.001 | <0.001 | <0.001 |
| Race | <0.001 | <0.001 | 0.006 | 0.045 |
| Visit Type | <0.001 | 0.004 | 0.033 | 0.172 |
| Study Outcome Group | 0.018 | 0.055 | 0.054 | 0.203 |
| Dog at Home | 0.037 | 0.106 | 0.683 | 0.838 |
| Daycare Attendance | 0.046 | 0.121 | 0.034 | 0.178 |
| Tobacco Exposure | 0.05 | 0.124 | 0.049 | 0.22 |
| Season | 0.968 | 0.979 | 0.006 | 0.045 |

further detail in the methods section. Briefly, we subsampled one sample per participant, performed PERMANVOA, and repeated this process 500 times to obtain an average $R^2$ and P-value. Using this method, most variables identified in the principal coordinate analysis remained significant, with dominant genus explaining the greatest proportion of variance in microbiota composition ($R^2 = 0.142$, P = 0.001; Canberra, Table 3). These findings were also largely significant when a Weighted UniFrac distance matrix was used, indicating that these environmental factors may influence the types of bacteria that dominate these communities.

## Compositionally distinct nasopharyngeal microbiota relates to URI and sinusitis

Because dominant bacterial genus explained a large proportion of variance, we postulated that compositionally and structurally distinct mucosal microbiota in the nasopharynx of these children and related to outcomes (i.e., uncomplicated URI or sinusitis). Using unsupervised hierarchical clustering, we identified four nasopharyngeal microbiota structures that differed significantly in bacterial composition (ANOVA; LME; p<0.001; Fig 1A). The selection of four clusters was supported by a comparison of average silhouette statistics (Fig 1B), where larger values represent greater separation between clusters. These compositionally distinct nasopharyngeal microbiota were characterized by being either *Moraxella*-dominated with sparse underlying taxonomy (Cluster I; n = 150); relatively diverse (Cluster II; n = 122); *Alloiococcus/ Corynebacterium* co-dominated (Cluster III; n = 65); or *Haemophilus*-dominated (Cluster IV;

**Table 3. Environmental, clinical, viral, and microbiological factors are associated with variance in bacterial community composition using a bootstrapped approach for repeated measures.**

| | Canberra[a] | | Weighted UniFrac[a] | |
| --- | --- | --- | --- | --- |
| Variable | $R^2$ | P-value | $R^2$ | P-value |
| Dominant Genus | 0.142 | 0.001 | 0.823 | 0.001 |
| Dominant Family | 0.120 | 0.001 | 0.814 | 0.001 |
| Study Outcome Group | 0.101 | 0.003 | 0.144 | 0.059 |
| Maternal Education | 0.096 | 0.024 | 0.098 | 0.321 |
| Visit Type | 0.085 | 0.003 | 0.227 | 0.001 |
| Race | 0.083 | 0.001 | 0.054 | 0.669 |
| Viral Detection | 0.022 | 0.024 | 0.059 | 0.022 |

[a]A single sample per individual is used to calculate PERMANOVA $R^2$ and p-values, repeating this process 500 times; the resulting mean $R^2$ and p-values are presented here.

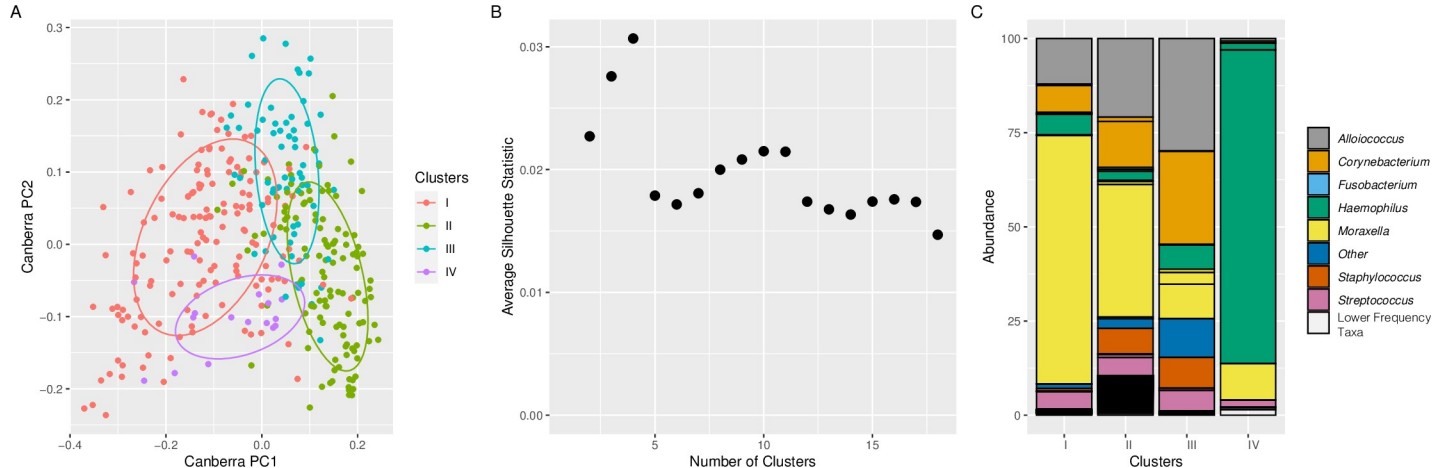

**Fig 1. Compositionally distinct nasopharyngeal microbiota are evident in longitudinally collected pediatric samples. A.** Four distinct microbiota assemblages are evident in nasopharyngeal samples of pediatric subjects [ANOVA(LME) P < 0.001, Canberra]. **B.** Average silhouette statistic supports the observation that a four-cluster solution best fits the model (greatest average silhouette statistic). **C.** Stacked bar plot of taxa from each cluster indicates that distinct bacterial distributions dominate each of the four nasopharyngeal microbiota. Each box represents a unique taxon within the cluster. Colors represent the genus-level identity of the taxon; white indicates lower frequency taxa.

n = 17; Fig 1C). Cluster II was significantly richer (Fig 2A), less dominated (Fig 2B), and more diverse (Fig 2C) compared to all other clusters, with enrichments of several underlying genera (S2 Fig).

Generalized estimating equations were used to determine whether between-cluster differences in clinical outcomes, virus detection, and participant-level (environmental and demographic) characteristics existed. Cluster I was associated with an increased risk for both sinusitis (RR = 1.25, P = 0.009; Table 3) and URI (RR = 1.18, P = 0.046), while Cluster II trended towards decreased risk of sinusitis (RR = 0.817; p = 0.08). Cluster II also exhibited a

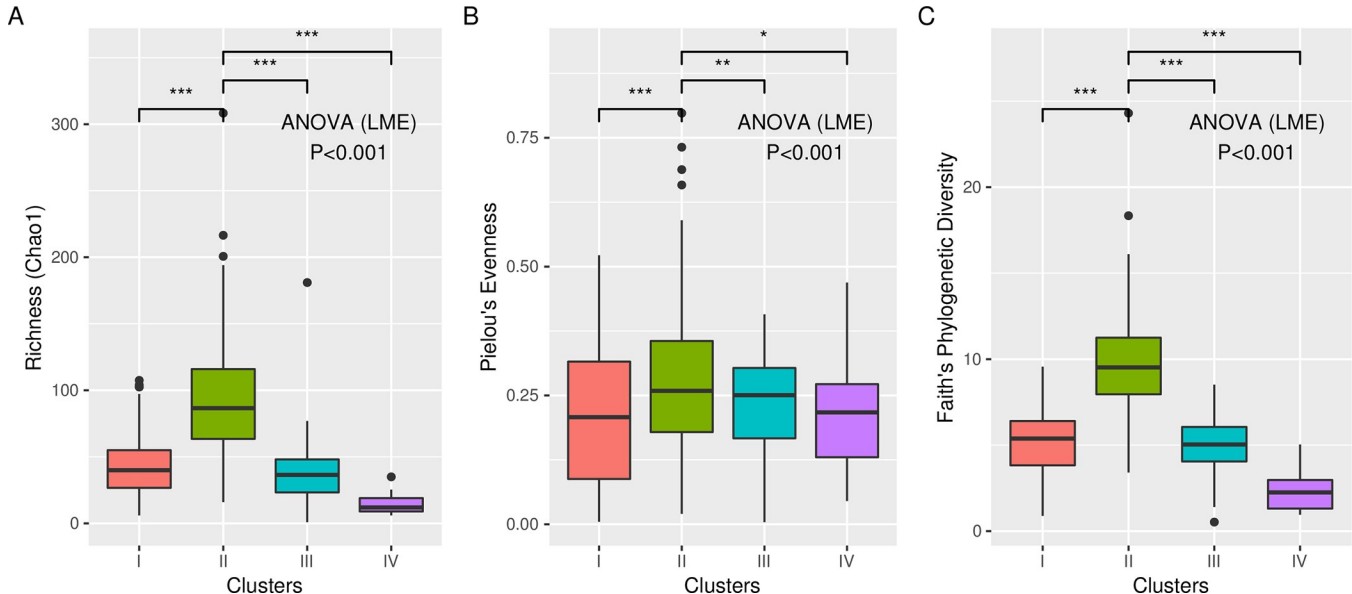

**Fig 2. Community structure is significantly different in nasopharyngeal microbiota.** Clusters are differentiated by **A.** richness, **B.** evenness and **C.** phylogenetic diversity, with cluster II exhibiting the greatest diversity across all three measures.

decrease in illness samples compared to all other clusters (RR = 0.847 P = 0.008), while Cluster IV was associated with a greater frequency of illness samples (RR = 1.056, P = 0.014). Cluster IV microbiota (n = 17) was also associated with reduced detection of RV-B and multiple RV (compared to samples with no virus; P = 0.001 for both) suggesting the increased illnesses observed in children with this nasopharyngeal microbiota were less frequently accompanied by RV infection. Nasopharyngeal microbiota also exhibited seasonal variation, with Cluster III more frequently detected in Fall and Summer and Cluster IV more prevalent in Spring and Winter (Table 4).

We additionally noted that about 75% (n = 110) of Cluster I samples were dominated by *Moraxella*, as were 50% (n = 56) of Cluster II samples. When focusing on *Moraxella*-dominated communities, clusters I and II exhibited consistently opposing, though non-significant, relationships with the study outcome group (Healthy vs. Sinusitis. vs. URI). *Moraxella* dominated communities from Cluster I were slightly more likely to be related to URI (RR = 1.22, P = 0.16) and Sinusitis (RR = 1.2, P = 0.22), while communities from Cluster II were slightly less likely to be related to both outcomes (URI RR = 0.86, P = 0.21; Sinusitis RR = 0.87, P = 0.21), suggesting that the direction of effect is not driven solely by samples not dominated by *Moraxella*.

### Longitudinal patterns of nasopharyngeal microbiota colonization

These nasopharyngeal microbiota profiles were then examined for their variability over time throughout the study period and with co-occurring rhinovirus infection (Fig 3). Of note, few children maintained the same nasopharyngeal profile throughout the study, highlighting the heterogeneity over the course of the year-long study period for these children, especially those developing sinusitis.

### Several taxa are depleted in children who develop URI and sinusitis

We next identified taxa detected over the sampling period that related to URI and Sinusitis outcomes. Children who developed URI or Sinusitis outcomes were consistently depleted of several bacterial taxa, including *Prevotella*, *Acetobacteraceae*, and *Chryseobacterium* (FDR P < 0.05; following adjustment for covariates previously identified: maternal education, tobacco exposure, season, and daycare attendance; Fig 4), while a single *Moraxella* taxon (OTU 1990) was significantly enriched in children who subsequently developed URI. This suggests that several bacterial taxa are present and abundant across the study period in children who do not go on to develop upper respiratory infection or sinusitis. This corroborates the findings identified through cluster analysis, in which communities enriched in underlying taxa were associated with protection against Sinusitis and URI events.

### Discussion

This study provides important insights into the microbial environment associated with the development of URI and acute sinusitis. Specifically, children having a *Moraxella*-dominated nasopharyngeal microbiota are at increased risk of URI or sinusitis. In baseline samples two *Moraxella* taxa were enriched in children who went on to develop sinusitis, while specific members of *Haemophilus*, *Staphylococcus* and *Streptococcus* were more abundant in children who did not go on to develop an upper respiratory infection. A previous study of the nasopharyngeal microbiota from this cohort, using a phylogenetic microarray, focused exclusively on initial samples collected from each subject. It determined that a history of sinusitis was associated with a significantly altered nasopharyngeal microbiota and enrichment of *Moraxella nonliquefaciens*, which also co-associated with subsequent development of sinusitis [4]. This

**Table 4. Microbiological, viral and clinical factors are differentially associated with microbiota assemblages.**

| | Cluster I n = 150 | | Cluster II n = 122 | | Cluster III n = 65 | | Cluster IV n = 17 | |
|---|---|---|---|---|---|---|---|---|
| | RR | P-value | RR | P-value | RR | P-value | RR | P-value |
| *Clinical Outcomes* | | | | | | | | |
| Study Outcome Group | | | | | | | | |
| Healthy (n = 31) | Reference | | Reference | | Reference | | Reference | |
| Sinusitis (n = 143) | 1.245 | 0.009 | 0.817 | 0.085 | 0.968 | 0.728 | 1.004 | 0.91 |
| URI (n = 180) | 1.18 | 0.046 | 0.839 | 0.153 | 0.971 | 0.757 | 1.03 | 0.411 |
| Visit Type | | | | | | | | |
| Surveillance/Entry (n = 141) | Reference | | Reference | | Reference | | Reference | |
| Sick (30 days; n = 8) | 0.753 | 0.036 | 1.242 | 0.226 | 1.123 | 0.46 | 0.971 | 0.012 |
| Sick (Acute; n = 98) | 1.103 | 0.162 | 0.818 | 0.001 | 1.053 | 0.363 | 1.043 | 0.081 |
| Sick (Acute—Sinusitis; n = 15) | 1.143 | 0.355 | 0.93 | 0.621 | 0.884 | 0.095 | 1.052 | 0.443 |
| Sick (Recovery; n = 79) | 0.943 | 0.347 | 1.014 | 0.833 | 1.002 | 0.974 | 1.032 | 0.25 |
| Sick (2nd Sinusitis; n = 13) | 1.222 | 0.179 | 0.782 | 0.028 | 0.846 | <0.001 | 1.235 | 0.075 |
| Visit Type (Simple) | | | | | | | | |
| Surveillance/Entry (n = 141) | Reference | | Reference | | Reference | | Reference | |
| Sick (n = 134) | 1.094 | 0.194 | 0.847 | 0.008 | 1.014 | 0.769 | 1.056 | 0.014 |
| Recovery (n = 79) | 0.941 | 0.336 | 1.016 | 0.802 | 0.999 | 0.981 | 1.033 | 0.25 |
| *Viral Outcomes* | | | | | | | | |
| Viral Detection | | | | | | | | |
| No (n = 165) | Reference | | Reference | | Reference | | Reference | |
| Yes (n = 189) | 1.081 | 0.156 | 0.93 | 0.141 | 1.001 | 0.982 | 0.999 | 0.956 |
| Total Viruses in Sample | 1.06 | 0.259 | 0.923 | 0.046 | 0.995 | 0.871 | 1.035 | 0.197 |
| HRV Type | | | | | | | | |
| None (n = 225) | Reference | | Reference | | Reference | | Reference | |
| HRV-A (n = 61) | 1.077 | 0.252 | 0.891 | 0.047 | 1.013 | 0.844 | 1.037 | 0.336 |
| HRV-B (n = 14) | 1.175 | 0.284 | 0.837 | 0.113 | 1.1 | 0.445 | 0.952 | 0.001 |
| HRV-C (n = 49) | 1.089 | 0.313 | 0.982 | 0.814 | 0.964 | 0.525 | 0.972 | 0.308 |
| HRV-M (n = 5) | 0.856 | 0.39 | 1.223 | 0.41 | 0.987 | 0.951 | 0.949 | 0.001 |
| *Participant Characteristics* | | | | | | | | |
| Season | | | | | | | | |
| Fall (n = 98) | Reference | | Reference | | Reference | | Reference | |
| Winter (n = 126) | 1.027 | 0.718 | 1.032 | 0.649 | 0.891 | 0.038 | 1.057 | 0.005 |
| Spring (n = 95) | 0.95 | 0.475 | 1.078 | 0.363 | 0.889 | 0.021 | 1.1 | 0.002 |
| Summer (n = 35) | 1.031 | 0.722 | 0.967 | 0.711 | 0.975 | 0.705 | 1.029 | 0.264 |

Models are generalized estimating equations with an exchangeable correlation structure, and Participant ID as the random effect.

report builds on our previous study by using 16S rRNA sequencing and longitudinally collected nasopharyngeal samples to define the characteristics of upper respiratory microbiota over the course of a year-long sampling period. The data reported in this study further supports earlier findings that a nasopharyngeal microbiota containing a broader range of bacterial phylogeny promotes protection against respiratory events in these children.

We related several variables to nasopharyngeal composition. While environmental exposures such as maternal education associated with variance in microbiota composition, the variance explained was relatively small, suggesting that a broad range of exogenous exposures exert small effects on established upper airway microbiota.

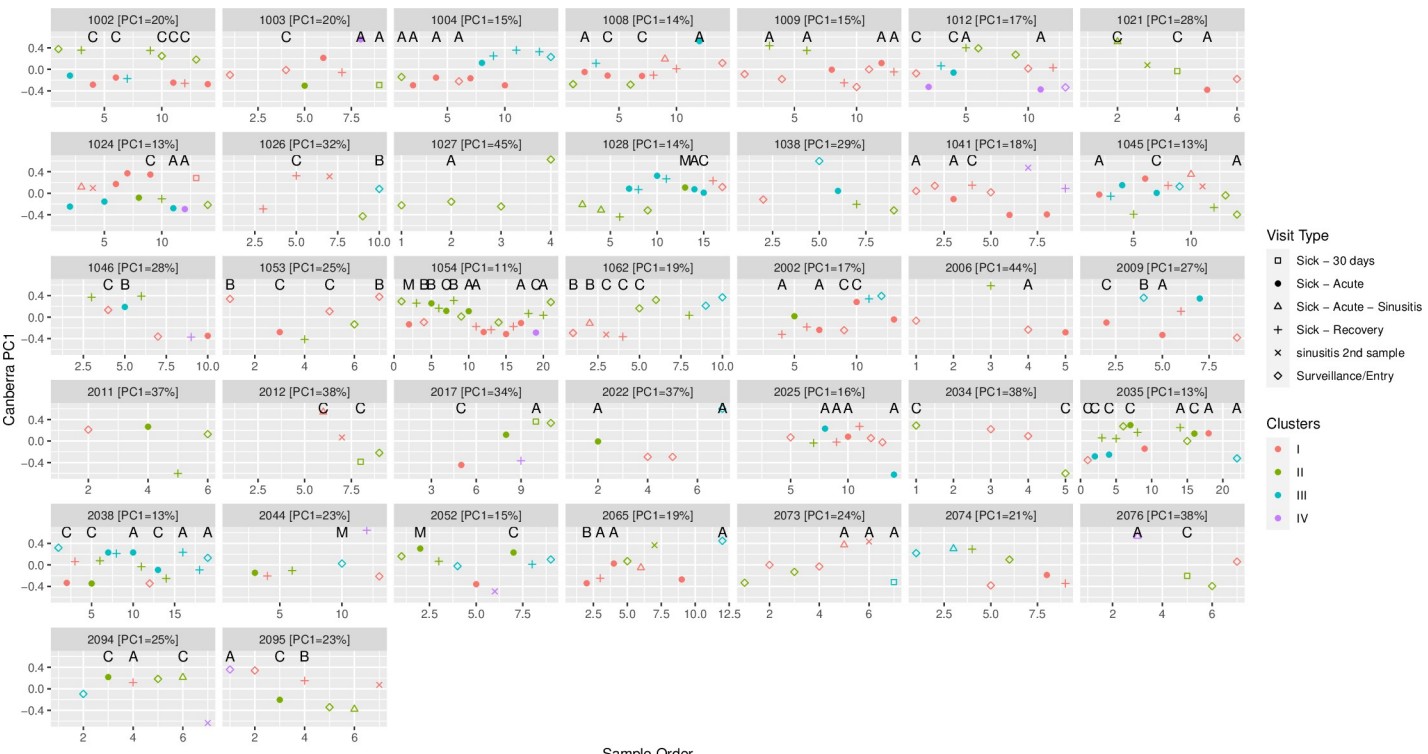

**Fig 3. Temporal dynamics of nasopharyngeal microbiota during health and upper respiratory illness.** Distances were re-ordinated for each person, and the proportion of variance explained by PC1 for each individual is displayed in the participant header. Data for individual participants is presented if they provided four or more samples over the course of the study. Letters represent the type of rhinovirus infection, if any. The four colors indicate the nasopharyngeal clusters identified, and the shape of the point denotes the detailed visit type in which the sample was obtained. Abbreviations (PC: Principal Coordinate; A: Rhinovirus A; B: Rhinovirus B; C: Rhinovirus C; M: Mixed Rhinovirus).

In contrast, the dominant bacterial genus explained a large proportion of the compositional variance of the microbiota, suggesting that microbe-host interactions, particularly in perturbed nasopharyngeal microbiomes, may represent the most important influence shaping composition, activities and interactions with the airway mucosa. This is consistent with recent findings in the gastrointestinal tract demonstrating that the established niche-specific endogenous microbiome governs colonization capacity of exogenous microbes [20] and suggests that pathogenic microbiomes may competitively exclude other species from the niche.

Cluster I microbiota, dominated primarily by *Moraxella*, exhibited lower microbiological richness and diversity, and associated with increased risk of sinusitis and URI. *Moraxella*, and more specifically *Moraxella catarrhalis*, is a common upper respiratory pathogen encoding a range of virulence factors that promote epithelial destruction and adherence [21–24]. Cluster II, characterized by a diverse underlying microbiota composition, associated with protection against sinusitis and URI despite a proportion of these samples being dominated by *Moraxella*. This supports a recent finding from an independent pediatric cohort identifying a protective effect of *Moraxella*-dominated microbiota; in those cases microbiota richness was also higher [25]. There are several explanations for this; *Moraxella* in Clusters I and II could represent distinct strains, with cluster I strains possessing a larger repertoire of virulence factors. An alternative possibility is that genetically similar *Moraxella* strains exist in Clusters I and II, but that interactions with distinct underlying microbiota members (or relative lack thereof) influence *Moraxella* burden and behavior. This is highlighted by the finding that several taxa were

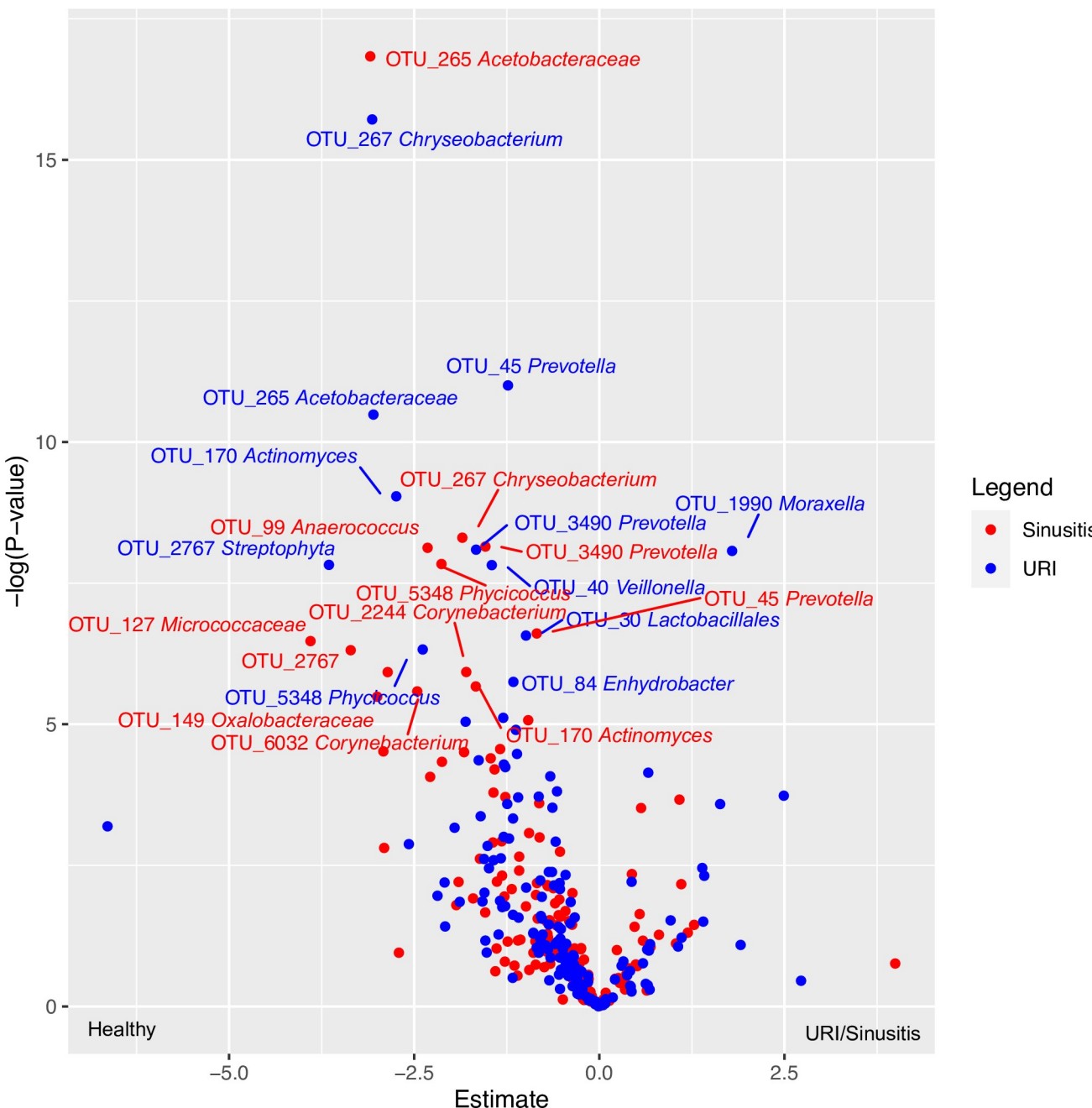

**Fig 4. Several taxa are relatively depleted in the nasopharyngeal microbiota of children who develop Upper Respiratory Infection (URI) and Sinusitis, while a specific Moraxella taxon (OTU 1990) is relatively enriched in children who go on to develop URI, but not Sinusitis.** Children who did not develop either URI or sinusitis are used as the comparison group. Red represents differential taxa in Sinusitis outcomes, while blue represents those taxa differential in URI outcomes (FDR p<0.05). Abbreviations: Upper Respiratory Infection (URI); Operational Taxonomic Unit (OTU); False Discovery Rate (FDR).

enriched in children who did not progress to URI or Sinusitis. This suggests that underlying nasopharyngeal microbiota play a critical role in airway pathogenesis and downstream sequalae. This second possibility has clinical implications and suggests that antimicrobial treatment, which is known to acutely deplete bacterial diversity in the upper respiratory tract [26], may

enhance *Moraxella* pathogenicity in the nasopharynx and increase susceptibility to subsequent clinical respiratory events.

These findings are not without limitations. First, this study is primarily generalizable to populations very similar to Madison, WI, and may not be generalizable to climates with different seasonal variation, or children from more diverse populations. In addition, this study identified several potential confounders, and studies with larger sample sizes may elucidate these findings further. We also note that the available bacterial profiles across the year varied from person to person, and having more complete data could have allowed for a refined analysis.

Distinct microbiota exist in the nasopharynx and relate to risk of URI and sinusitis events. Nasopharyngeal microbiota dominated by *Moraxella* are associated with respiratory events. However, simply detecting *Moraxella* may be insufficient to predict these outcomes since this relationship appears dependent on the underlying composition of the microbiota. Thus, studies examining bacterial strain level-genomic differentials coupled with an examination of nasopharyngeal microbiota function in the context of respiratory outcomes is necessary to fully elucidate how the airway microbiome modulates respiratory health. Such studies would facilitate precision microbiome manipulation strategies to mitigate the development of sinusitis in susceptible pediatric populations.

## Supporting information

**S1 Fig. Graphic describing the study design and sample collection scheme, in which children were followed for one year for development of upper respiratory infection or sinusitis and quarterly well-visit samples were collected.**
(TIF)

**S2 Fig. Heatmap of nasopharyngeal clusters.** Taxa were agglomerated at the genus level, and abundances were log-transformed. The top 30 genera are presented in the heatmap.
(TIF)

## Author Contributions

**Conceptualization:** Gregory DeMuri, Nabeetha N. Nagalingam, Ellen R. Wald, Susan V. Lynch.

**Data curation:** Kathryn E. McCauley, Gregory DeMuri, Kole Lynch, Douglas W. Fadrosh, Clark Santee, Nabeetha N. Nagalingam, Ellen R. Wald, Susan V. Lynch.

**Funding acquisition:** Gregory DeMuri, Ellen R. Wald, Susan V. Lynch.

**Investigation:** Clark Santee, Ellen R. Wald, Susan V. Lynch.

**Methodology:** Kathryn E. McCauley, Ellen R. Wald.

**Project administration:** Ellen R. Wald.

**Software:** Kathryn E. McCauley.

**Supervision:** Susan V. Lynch.

**Visualization:** Kathryn E. McCauley.

**Writing – original draft:** Gregory DeMuri, Ellen R. Wald, Susan V. Lynch.

**Writing – review & editing:** Kathryn E. McCauley, Gregory DeMuri, Ellen R. Wald, Susan V. Lynch.

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
