## [Decision Letter · Decision Letter 0]

24 May 2021

PONE-D-21-07837

Moraxella-dominated Pediatric Nasopharyngeal Microbiota Associate with Upper Respiratory Infection and Sinusitis

PLOS ONE

Dear Dr. Lynch,

Thank you for submitting your manuscript to PLOS ONE. After careful consideration, we feel that it has merit but does not fully meet PLOS ONE’s publication criteria as it currently stands. Therefore, we invite you to submit a revised version of the manuscript that addresses the points raised during the review process.

The reviewers raise a number of points that require consideration, particularly those related to methodological aspects. 

Please submit your revised manuscript by Jul 08 2021 11:59PM. If you will need more time than this to complete your revisions, please reply to this message or contact the journal office at plosone@plos.org. Please include the following items when submitting your revised manuscript:

We look forward to receiving your revised manuscript.

Kind regards,

Aran Singanayagam

Academic Editor

PLOS ONE

Journal Requirements:

2)  You indicated that you had ethical approval for your study. In your Methods section, please ensure you have also stated whether you obtained consent from parents or guardians of the minors included in the study or whether the research ethics committee or IRB specifically waived the need for their consent.

3) In your Methods section, please provide additional information about the participant recruitment method and the demographic details of your participants. Please ensure you have provided sufficient details to replicate the analyses such as: a) the recruitment date range (month and year), b) a table of relevant demographic details, c) a description of how participants were recruited, and d) descriptions of where participants were recruited and where the research took place.

4) Please provide a sample size and power calculation in the Methods, or discuss the reasons for not performing one before study initiation.

5) Please ensure you have discussed any potential limitations of your study in the Discussion, including study design, sample size and/or potential confounders.

6) PLOS requires an ORCID iD for the corresponding author in Editorial Manager on papers submitted after December 6th, 2016. Please ensure that you have an ORCID iD and that it is validated in Editorial Manager. To do this, go to ‘Update my Information’ (in the upper left-hand corner of the main menu), and click on the Fetch/Validate link next to the ORCID field. This will take you to the ORCID site and allow you to create a new iD or authenticate a pre-existing iD in Editorial Manager. Please see the following video for instructions on linking an ORCID iD to your Editorial Manager account: https://www.youtube.com/watch?v=_xcclfuvtxQ

7) Please include captions for your Supporting Information files at the end of your manuscript, and update any in-text citations to match accordingly. Please see our Supporting Information guidelines for more information: http://journals.plos.org/plosone/s/supporting-information.

8) We note that you have a patent relating to material pertinent to this article. Please provide an amended statement of Competing Interests to declare this patent (with details including name and number), along with any other relevant declarations relating to employment, consultancy, patents, products in development or modified products etc. Please confirm that this does not alter your adherence to all PLOS ONE policies on sharing data and materials, as detailed online in our guide for authors http://journals.plos.org/plosone/s/competing-interests by including the following statement: "This does not alter our adherence to  PLOS ONE policies on sharing data and materials.” If there are restrictions on sharing of data and/or materials, please state these. Please note that we cannot proceed with consideration of your article until this information has been declared.

Reviewers' comments:

Reviewer's Responses to Questions

**Comments to the Author**

1. Is the manuscript technically sound, and do the data support the conclusions?

Reviewer #1: Partly

Reviewer #2: Yes

2. Has the statistical analysis been performed appropriately and rigorously? 

Reviewer #1: No

Reviewer #2: I Don't Know

3. Have the authors made all data underlying the findings in their manuscript fully available?

Reviewer #1: No

Reviewer #2: Yes

4. Is the manuscript presented in an intelligible fashion and written in standard English?

Reviewer #1: Yes

Reviewer #2: Yes

5. Review Comments to the Author

Reviewer #1: Minor comments:

1. The authors mention in the methods part that they include negative controls. However, they don't mention what these controls actually were and how were they collected.

2. It is not clear if the authors performed 16S rRNA gene qPCR on the samples (no methods part) and thereby used the term "copy numbers" ? If not copy number cannot be used as equivalent for reads.

3. Line 130-131: "103 did not produce sufficient DNA for 16S rRNA amplification or failed to produce an amplicon", how was this checked and what is the threshold ?

4. Line 180-181: Please explain what dominant genera here means ? If the authors are talking about frequency of presence that would be prevalence. Dominance would be defined as the most abundant taxa in a sample.

5. The table in the PDF is not properly visible and poorly presented.

Major comments:

1. Please submit a proper script with instructions, metadata table on the GitHub page mentioned in the manuscript. The authors may also use Zenodo. It is evident that the authors used QIIME but compiling the script with modifications to native scripts, if any would be a great addition. The authors do provide them in the Data availability section but please add it to methods at relevant places or cite the GitHub page.

2. Why Canberra distance ? Why not Bray-Curtis? Compositional data has two inherent factors that contribute to the ecosystem. 1. Presence / Absence of a species 2. Abundance of a species. BC distance is most common since it combines both of these metrics. Having said this, if the authors still wish to use this metric then please explain the rationale behind it?

3. If the authors have indeed performed 16S rRNA copy number with qPCR on all samples, please provide a histogram plot showing the range of copy numbers by samples.

4. The starting of the results should include general characteristics of the nasopharyngeal microbiota in terms how many OTUs were obtained and how many genera. May be adding a stacked bar plot with time.

5. Metadata parameters:

The authors do not explain the rationale behind the metadata and how they are collected. Please provide more details in the methods. Some of these metadata do not seem to have any relationship or strong hypothesis. If this was intended then is a fundamental flaws in study design to introduce certain factors that the authors cannot check for with such a small sample size. Having metadata doesn't mean that it should used for PERMANOVA, if it cannot explain the data well.

M. catarrhalis Copy Number (log) - was this calculated by qPCR ? If not the authors cannot single out one species. The clusters containing this bacteria differentiate by the applied statistics. Hence, this is unnecessary and provides no extra information than the M. catarrhalis dominated cluster.

Dominant Genus - Not sure what this means, please explain.

H. influenzae Copy Number (log) - Same issue as above.

S. pneumoniae Copy Number (log) - Same issue as above

Season - how is this defined here ? Precisely which time of the year is it and how far apart are the samplings througout one season ?

Daycare Attendance - Is it just attendance ? yes or no but at what time and how was this factor controlled for?

Tobacco Exposure - Again what does actually mean ? Passive smoking ? or any oral tobacco? How can the authors control for this ?

Study Sub-Group - Its either the same as the groups below, i.e. Healthy vs Sick or the other sub-groups. Again a nested factor and I am not sure if this is actually necessary as all will say practically the same thing.

Health vs Sick Individuals - This is a more generalised description and a nested factor for the below mentioned sub-groups.

Surveillance vs. Sick vs. Recovery Sample Sinusitis vs. Non-Sinusitis Sample Number of Surveillance Visits

Visit Type - same issue as above

Mother's Age - what is the relationship with 16S data ?

Mother's Education - This is absurd! Please explain what do the author's mean by this ?

Other Children - Not sure what this means ?

Total Virus Types in Sample - What metric is used for this ?

Child Shares a Room - How does this matter when children can interaction outside the room as well. Is this factor controlled ?

Race - Do the authors imply that they exclude mixed race children from the study ? How do the authors define race ? Was there any genetic test done or this was just word of mouth ? Race-based analyses of microbiota do exist but this study doesn't have enough statistical power in terms of sample numbers to really test this.

Size of Household - Not a relevant factor.

Dog at Home - What is the hypothesis ?

Dog at School - What is the hypothesis ?

Cat at Home - What is the hypothesis ?

Cat at School - What is the hypothesis ?

Ethnicity - Same problem with race. What has ethinicity to do with any of the biological questions ? How does one judge ethnicity? Is there enough sample size to explore this ?

6. The authors mentioned the usage of UniFrac distance for first result (line 198-200). UniFrac is phylogenetic distance-based metric, which requires a phylogenetic tree. However, the authors do not mention in the methods as to how they made the tree.

7. Why use UniFrac for the first result but Canberra distance for the clustering ? What is the rationale here?

8. Line 202: What do the authors mean by " based on species-specific copy number" ? This relates to my concern raised in Point number 5.

9. Please add plot showing change in dynamics over time.

10. Why do the authors use faith's diversity for the plot when they have weighted UniFrac ?

11. Figure 3 is unreadable and if its coming out of a DESeq2 analysis, please provide volcano plots instead.

12. Line 275-280: The authors explain the phenomenon here but unfortunately this is not visible in a plot or graph of some sort. Please make it easier for the reader.

13. The authors use DESeq2 but didn't mention the model formula used ? What were the factors used ? The authors do mention in the figure that the factors are healthy vs disease but please mention this in the methods. Please also mention if the model used was a one-factorial or two-factorial design i.e. Time being another factor here.

14. Continuing with the previous point, the authors do not take sufficient advantage of the longitudinal data they have. Neither do they show movement of the clusters over time nor do they perform differential taxa analysis over time.

15. If this study goes on to claim anything the analysis has to be rock solid and it isn't now. One way to improve quality is to use DADA2 based filtering before OTU assignment and classification. The authors can use this or provide an argument against it.

Reviewer #2: This study is focused on understanding the longitudinal change in nasopharyngeal microbiota of children and its association with development of upper respiratory tract infections (URI) or sinusitits. The same group has previous published an observational study and proved that children with Moraxella dominated microbiota are at high risk of infection. The novelty of this study is here the samples were collected from healthy children and they are followed up for a period of 3 years and samples were collected periodically. This study also revealed the association of Moraxella at baseline with URI infection on subsequent visits. The study is well planned and appropriate analyses were performed. The following queries are need to be clarified.

1. Are the children are healthy controls or they visited the pediatric centre for other health problems like fever, diahorrea, etc., In case, if they are healthy, what is the purpose of their visit to pediatric centre? Are they invited specifically for this study?

2. A figure describing the timeline of sample collection and methodology will enable easy understanding of the longitudinal sample collection.

3. For viral identification, a panel of viruses were detected by multiplex PCR. The list includes both DNA and RNA viruses. Does RNA isolation and cDNA construction is performed for detecting RNA viruses? Clarify

4. In methods, under sub-heading – procedures, it is mentioned nasal samples were obtained. The authors should elaborate how nasal samples are collected?

5. Though the authors studied the presence of differet viruses by multiplex PCR, the results are not correlated with the bacterial composition. It will be interesting to find any association between a specific viral type and a dominant bacteria.

6. Table 3 is not fully visible. Footnotes onbrief details of statistical analysis can be added under the tables.

6. PLOS authors have the option to publish the peer review history of their article (what does this mean?). If published, this will include your full peer review and any attached files.

Reviewer #1: No

Reviewer #2: **Yes: **Ganesan Velmurugan

---

## [Author Response · Author response to Decision Letter 0]

5 Nov 2021

Reviewer #1: Minor comments:

1. The authors mention in the methods part that they include negative controls. However, they don't mention what these controls actually were and how were they collected.

We thank you for this comment. The negative controls were phosphate-buffered saline and underwent the same technical processing as samples. These negative controls were then used, as described in the methods, to remove technical background contamination from samples.

2. It is not clear if the authors performed 16S rRNA gene qPCR on the samples (no methods part) and thereby used the term "copy numbers" ? If not copy number cannot be used as equivalent for reads.

Given our agreement that inclusion of qPCR values do not add to the overall results, we have removed the qPCR findings and all description of those calculations.

3. Line 130-131: "103 did not produce sufficient DNA for 16S rRNA amplification or failed to produce an amplicon", how was this checked and what is the threshold ?

We use the Qubit HS dsDNA kit from Invitrogen, and we aim for more than 2 ng/uL. This information has been added to the methods.

4. Line 180-181: Please explain what dominant genera here means ? If the authors are talking about frequency of presence that would be prevalence. Dominance would be defined as the most abundant taxa in a sample.

We thank the reviewer for this comment and have clarified the definition of dominant genus accordingly.

5. The table in the PDF is not properly visible and poorly presented.

We apologize for the lack of clarity of Table 3. We were following the guidance for tables where it states “Do not split your table or otherwise try to make the table appear within the manuscript margins if it does not fit on one page. In Word, tables that run off of the manuscript page can be seen using Draft View”. (https://journals.plos.org/plosone/s/tables). The table has undergone edits to improve clarity.

Major comments:

1. Please submit a proper script with instructions, metadata table on the GitHub page mentioned in the manuscript. The authors may also use Zenodo. It is evident that the authors used QIIME but compiling the script with modifications to native scripts, if any would be a great addition. The authors do provide them in the Data availability section but please add it to methods at relevant places or cite the GitHub page.

We appreciate the interest in obtaining additional documentation about our statistical procedures. Code that generated Tables and Figures are available in the github repository for this study, or as code in other repositories, now clearly identified in the methods.

2. Why Canberra distance ? Why not Bray-Curtis? Compositional data has two inherent factors that contribute to the ecosystem. 1. Presence / Absence of a species 2. Abundance of a species. BC distance is most common since it combines both of these metrics. Having said this, if the authors still wish to use this metric then please explain the rationale behind it?

We thank the reviewer for this feedback. We used all four distance matrices in an exploratory manner to identify microbiota clusters. While Canberra may have not provided the highest silhouette statistic, it produces statistically significant clinically-relevant findings. We have now expanded upon this finding in our manuscript. 

3. If the authors have indeed performed 16S rRNA copy number with qPCR on all samples, please provide a histogram plot showing the range of copy numbers by samples.

Given our agreement that inclusion of qPCR values do not add to the overall results, we have removed the qPCR findings and all description of those calculations.

4. The starting of the results should include general characteristics of the nasopharyngeal microbiota in terms how many OTUs were obtained and how many genera. May be adding a stacked bar plot with time.

We agree and have now added information about the general characteristics of these nasopharyngeal microbiota to the first paragraph of the results.

5. Metadata parameters:

The authors do not explain the rationale behind the metadata and how they are collected. Please provide more details in the methods. Some of these metadata do not seem to have any relationship or strong hypothesis. If this was intended then is a fundamental flaws in study design to introduce certain factors that the authors cannot check for with such a small sample size. Having metadata doesn't mean that it should used for PERMANOVA, if it cannot explain the data well.

We thank you for this comment, as such we have reduced Table 2 to the specific significant factors in our analysis, while maintaining the FDR p-value so that readers can interpret our findings knowing that other factors were tested but were not found to be significant.

M. catarrhalis Copy Number (log) - was this calculated by qPCR ? If not the authors cannot single out one species. The clusters containing this bacteria differentiate by the applied statistics. Hence, this is unnecessary and provides no extra information than the M. catarrhalis dominated cluster.

We agree that including qPCR copy number of common respiratory pathogens does not enhance the interpretation of the 16S sequencing data, and thus we have removed these variables from consideration. 

Dominant Genus - Not sure what this means, please explain.

We have now explained this variable in greater detail in the methods.

H. influenzae Copy Number (log) - Same issue as above.

S. pneumoniae Copy Number (log) - Same issue as above

Season - how is this defined here ? Precisely which time of the year is it and how far apart are the samplings througout one season ?

We have included a more complete definition of season in the methods.

Daycare Attendance - Is it just attendance ? yes or no but at what time and how was this factor controlled for?

We have included a more complete definition of daycare attendance in the methods which should address this concern.

Tobacco Exposure - Again what does actually mean ? Passive smoking ? or any oral tobacco? How can the authors control for this ?

For clarity, we have expanded the methods to include the specific question asked of the parents during the extensive questionnaire session.

Study Sub-Group - Its either the same as the groups below, i.e. Healthy vs Sick or the other sub-groups. Again a nested factor and I am not sure if this is actually necessary as all will say practically the same thing.

Agreed – this variable was created to allow for the inclusion of samples within groups with very small sample sizes (ie, 30-day illness samples of n=8, or second Sinusitis samples of n=13). More information has been included in the methods, and most nested factor variables have now been removed.

Health vs Sick Individuals - This is a more generalised description and a nested factor for the below mentioned sub-groups.

Surveillance vs. Sick vs. Recovery Sample Sinusitis vs. Non-Sinusitis Sample Number of Surveillance Visits

Visit Type - same issue as above

Mother's Age - what is the relationship with 16S data ?

This variable has been removed from the set of reported variables.

Mother's Education - This is absurd! Please explain what do the author's mean by this ?

We respectfully note that maternal education often represents differences in socioeconomic status, which can account for several measured and unmeasured confounders. Our study uses maternal education, in addition to other significant factors, as a confounder in our downstream analyses.

Other Children - Not sure what this means ?

We have clarified this to state “Other Children in the Household” and again have expanded the methods to include the specific question being asked in the questionnaire. Nasopharyngeal microbiome has been shown to relate to the number of other children in the household, plausibly due to microbial sharing amongst family members.

Total Virus Types in Sample - What metric is used for this ?

This is now explained in the methods section.

Child Shares a Room - How does this matter when children can interaction outside the room as well. Is this factor controlled ?

Agreed that the child can interact with others outside of the shared room, but room sharing is important since it indicates a high-degree of daily shared environment for the child. 

Race - Do the authors imply that they exclude mixed race children from the study ? How do the authors define race ? Was there any genetic test done or this was just word of mouth ? Race-based analyses of microbiota do exist but this study doesn't have enough statistical power in terms of sample numbers to really test this.

As now noted in the methods, race was ascertained through self-report. Mixed-race participants were not excluded from the study. 

Size of Household - Not a relevant factor.

Dog at Home - What is the hypothesis ?

Dog at School - What is the hypothesis ?

Cat at Home - What is the hypothesis ?

Cat at School - What is the hypothesis ?

Ethnicity - Same problem with race. What has ethinicity to do with any of the biological questions ? How does one judge ethnicity? Is there enough sample size to explore this ?

We appreciate the reviewer’s interest in how these variables were ascertained. As such, we have expanded the methods considerably to help provide more clarity. These factors are included in our study because they have been shown in many large epidemiological studies to relate to airway disease development and prevalence. Hence, we determined in our study whether these pre-existing relationships between these variables and airway disease, could be explained by relationships with the upper airway microbiota. We univariately related these variables to upper respiratory microbiota to understand factors that shape colonization patterns in these young children, and respectfully believe this is a fair approach to take. We also respectfully believe that factors such as maternal education may indirectly relate to the child’s upper respiratory microbiota, as it is an indicator of socioeconomic status, and thus can influence upper respiratory microbiota through differences in pollution, diet and stress.

6. The authors mentioned the usage of UniFrac distance for first result (line 198-200). UniFrac is phylogenetic distance-based metric, which requires a phylogenetic tree. However, the authors do not mention in the methods as to how they made the tree.

We appreciate this notification and have included the methods used to construct a phylogenetic tree and further refine the taxa included in our dataset.

7. Why use UniFrac for the first result but Canberra distance for the clustering ? What is the rationale here?

As mentioned above, several distance matrices were used in our study, since each weights different aspects of microbial community composition and significant findings based on any one of these matrices offers insights into the nature of the relationship. We thus present those findings that produce the strongest data and indicate in our narrative the interpretation of these findings based on different distance matrices. 

8. Line 202: What do the authors mean by " based on species-specific copy number" ? This relates to my concern raised in Point number 5.

Given our agreement that inclusion of qPCR values do not add to the overall results, we have removed the qPCR findings and all description of those calculations.

9. Please add plot showing change in dynamics over time.

We appreciate this comment and have included a figure (Figure 3) of individual composition changes over time, their microbiota cluster, and concurrent rhinovirus infection.

10. Why do the authors use faith's diversity for the plot when they have weighted UniFrac ?

We plotted three alpha diversity metrics in Figure 2. Beta Diversity metrics are presented in Tables 2 and 3.

11. Figure 3 is unreadable and if its coming out of a DESeq2 analysis, please provide volcano plots instead.

Thank you for this comment – Figure 3 has been replaced by a volcano plot displaying the differential taxa, utilizing a new method that allows for the repeated samplings to be considered with mixed effects models. This is described further in the methods section and the R script is publicly available on the cited GitHub repository.

12. Line 275-280: The authors explain the phenomenon here but unfortunately this is not visible in a plot or graph of some sort. Please make it easier for the reader.

Per several comments, the data associated with Figure 3 have been generated with a new statistical method which show slightly distinct findings, though consistent with the narrative. Additionally, data are now presented in a volcano plot to increase interpretability.

13. The authors use DESeq2 but didn't mention the model formula used ? What were the factors used ? The authors do mention in the figure that the factors are healthy vs disease but please mention this in the methods. Please also mention if the model used was a one-factorial or two-factorial design i.e. Time being another factor here.

We appreciate these additional questions about the specifics of our statistical model. We used DESeq2 on cross-sectional data, as DESeq2 doesn’t include methods to effectively handle repeated measures. Since this manuscript was submitted, we have applied a newly developed script to the data, available at our github and cited in the methods. This script applies several mixed effects models and determines which fits the data best using the Aikake Information Criterion, and reports the best-fitting resulting estimate and p-value. In addition, not unlike DESeq2, multivariable models can be utilized, and we have used the information regarding covariates relating to upper respiratory microbiomes to adjust our models, identifying taxa that relate to URI and sinusitis after adjusting for confounders such as maternal education.

14. Continuing with the previous point, the authors do not take sufficient advantage of the longitudinal data they have. Neither do they show movement of the clusters over time nor do they perform differential taxa analysis over time.

We appreciate the interest in the longitudinal data collected for this study. Per a previous comment, we have now included movement of the clusters over time and its relationship with community composition. In addition, the taxonomic analysis now utilizes the repeated measures available. 

15. If this study goes on to claim anything the analysis has to be rock solid and it isn't now. One way to improve quality is to use DADA2 based filtering before OTU assignment and classification. The authors can use this or provide an argument against it.

We thank the reviewer for this comment and believe it has helped us improve the quality of the analyses we presented in this manuscript. This analysis uses stringent quality metrics before OTU assignment, which include trimming sequences with three or more consecutive bases with a Q-score less than 30. For this study, we chose to move forward with an OTU/USEARCH-based method to help reduce dimensionality of the dataset.

Reviewer #2: This study is focused on understanding the longitudinal change in nasopharyngeal microbiota of children and its association with development of upper respiratory tract infections (URI) or sinusitits. The same group has previous published an observational study and proved that children with Moraxella dominated microbiota are at high risk of infection. The novelty of this study is here the samples were collected from healthy children and they are followed up for a period of 3 years and samples were collected periodically. This study also revealed the association of Moraxella at baseline with URI infection on subsequent visits. The study is well planned and appropriate analyses were performed. The following queries are need to be clarified.

1. Are the children are healthy controls or they visited the pediatric centre for other health problems like fever, diahorrea, etc., In case, if they are healthy, what is the purpose of their visit to pediatric centre? Are they invited specifically for this study?

We thank you for this clarification request. Children were enrolled from their routine primary care clinics and recruited for the study during well-child visits, so they were typically asymptomatic at the time of recruitment. In addition, several exclusion criteria were used for the study and these are provided in the manuscript. We have included this clarification in the methods.

2. A figure describing the timeline of sample collection and methodology will enable easy understanding of the longitudinal sample collection.

We agree, and as such have included a study sampling schematic as well as an analysis of the nasopharyngeal composition over time in these children.

3. For viral identification, a panel of viruses were detected by multiplex PCR. The list includes both DNA and RNA viruses. Does RNA isolation and cDNA construction is performed for detecting RNA viruses? Clarify

We thank you for this request for clarification. Our specimen processing extracts both RNA and DNA and the multiplex chemistry includes an RT step. Therefore, we are likely detecting viruses from both cDNA and genomic DNA.

4. In methods, under sub-heading – procedures, it is mentioned nasal samples were obtained. The authors should elaborate how nasal samples are collected?

This information was previously included under the heading “Collection of samples”, in which we describe that a flocked swab was placed into the nasopharynx and rotated, and samples were stored in sterile DNAase/RNAase-free cryovials of RNAlater. For clarity, we have moved the description of sample collection to the Procedures section.

5. Though the authors studied the presence of different viruses by multiplex PCR, the results are not correlated with the bacterial composition. It will be interesting to find any association between a specific viral type and a dominant bacteria.

We appreciate this feedback and performed the associated analysis. We found that the dominant bacteria did not relate to the viral type. However, we do note a significant relationship between community composition and viral detection in Table 3.

6. Table 3 is not fully visible. Footnotes on brief details of statistical analysis can be added under the tables.

We apologize for the lack of clarity of Table 3. We were following the guidance for tables where it states “Do not split your table or otherwise try to make the table appear within the manuscript margins if it does not fit on one page. In Word, tables that run off of the manuscript page can be seen using Draft View”. (https://journals.plos.org/plosone/s/tables). We have made minor changes to the formatting of the table to improve visibility of the table within the PDF.

We have also included footnotes to provide additional information about the statistical methods used to obtain statistical relationships.

---

## [Decision Letter · Decision Letter 1]

29 Nov 2021

Moraxella-dominated Pediatric Nasopharyngeal Microbiota Associate with Upper Respiratory Infection and Sinusitis

PONE-D-21-07837R1

Dear Dr. Lynch,

We’re pleased to inform you that your manuscript has been judged scientifically suitable for publication and will be formally accepted for publication once it meets all outstanding technical requirements.

Kind regards,

Aran Singanayagam

Academic Editor

PLOS ONE

Additional Editor Comments (optional):

Reviewers' comments:

Reviewer's Responses to Questions

**Comments to the Author**

1. If the authors have adequately addressed your comments raised in a previous round of review and you feel that this manuscript is now acceptable for publication, you may indicate that here to bypass the “Comments to the Author” section, enter your conflict of interest statement in the “Confidential to Editor” section, and submit your "Accept" recommendation.

Reviewer #1: All comments have been addressed

Reviewer #2: (No Response)

2. Is the manuscript technically sound, and do the data support the conclusions?

Reviewer #1: Yes

Reviewer #2: Yes

3. Has the statistical analysis been performed appropriately and rigorously? 

Reviewer #1: Yes

Reviewer #2: Yes

4. Have the authors made all data underlying the findings in their manuscript fully available?

Reviewer #1: Yes

Reviewer #2: Yes

5. Is the manuscript presented in an intelligible fashion and written in standard English?

Reviewer #1: Yes

Reviewer #2: Yes

6. Review Comments to the Author

Reviewer #1: The authors have put a great effort to diligently address my concerns and provided the necessary data. This has remarkably improved the manuscript.

Reviewer #2: (No Response)

7. PLOS authors have the option to publish the peer review history of their article (what does this mean?). If published, this will include your full peer review and any attached files.

Reviewer #1: **Yes: **Sudip Das, University of Lausanne, Switzerland

Reviewer #2: **Yes: **Velmurugan Ganesan

---

## [Editor Report · Acceptance letter]

16 Dec 2021

PONE-D-21-07837R1 

*Moraxella*-dominated Pediatric Nasopharyngeal Microbiota Associate with Upper Respiratory Infection and Sinusitis 

Dear Dr. Lynch:

I'm pleased to inform you that your manuscript has been deemed suitable for publication in PLOS ONE. Congratulations! Your manuscript is now with our production department. 

Kind regards, 

on behalf of

Dr. Aran Singanayagam 

Academic Editor

PLOS ONE